# An Outbreak of *Aeromonas salmonicida* in Juvenile Siberian Sturgeons (*Acipenser baerii*)

**DOI:** 10.3390/ani13172697

**Published:** 2023-08-23

**Authors:** Esther Vázquez-Fernández, Blanca Chinchilla, Agustín Rebollada-Merino, Lucas Domínguez, Antonio Rodríguez-Bertos

**Affiliations:** 1VISAVET Health Surveillance Centre, Complutense University of Madrid, 28040 Madrid, Spain; esthvazq@ucm.es (E.V.-F.); bchinchi@ucm.es (B.C.); agusrebo@ucm.es (A.R.-M.); lucasdo@ucm.es (L.D.); 2Department of Internal Medicine and Animal Surgery, Faculty of Veterinary Medicine, Complutense University of Madrid, 28040 Madrid, Spain; 3Department of Animal Health, Faculty of Veterinary Medicine, Complutense University of Madrid, 28040 Madrid, Spain

**Keywords:** Siberian sturgeon, *Aeromonas salmonicida*, macroscopic lesions, histopathology, aquaculture, MALDI-TOF

## Abstract

**Simple Summary:**

*Aeromonas salmonicida* represents an important cause of mortality in a broad diversity of fish species, in which it has been extensively reported. Nevertheless, aeromoniasis in sturgeons is far from being fully understood. Four juvenile sturgeons from a mortality outbreak in a farm located in the north of Spain were submitted to the VISAVET Health Surveillance Centre. *Aeromonas salmonicida* was isolated from the bacteriological culture. They were found to suffer from a hyperacute form of the disease (almost no clinical signs and non-specific gross or histological lesions), thus confirming the susceptibility of sturgeons to this pathogen. The findings described here hope to provide a better understanding of this potentially fatal disease in the industries of aquaculture and caviar.

**Abstract:**

*Aeromonas salmonicida* is one of the major threats to world aquaculture, causing fish furunculosis and high mortality rates in cultured fish, particularly salmonids. Although *Aeromonas* spp. is a thoroughly studied pathogen, little is known regarding aeromoniasis in sturgeons. After a mortality outbreak, four juvenile sturgeons (*Acipenser baerii*) were submitted for autopsy and tissue samples were collected for histopathological and microbiological studies. The external examination revealed size heterogenicity, skin hyperpigmentation and reduced body condition of sturgeons. Within the abdominal cavity, mild hepatomegaly and splenomegaly were observed, as well as generalized organic congestion. Histology revealed severe multifocal haemorrhagic and ulcerative dermatitis, mainly localized in the dorsal and latero-ventral areas of fish. The histological study also showed moderate to severe inflammation of gills and organic lesions compatible with septicaemia. Bacterial isolates were identified as *Aeromonas salmonicida* subsp. *salmonicida* using MALDI-TOF MS and PCR. Overall, the lesions first described here are consistent with those previously reported in other cultured fish species and contribute to a better understanding of the pathogenesis of *Aeromonas salmonicida* subsp. *salmonicida* in the Siberian sturgeon, aside from providing new diagnostic tools for bacterial diseases impacting the fast-growing industry of caviar.

## 1. Introduction

Sturgeon farming has acquired an increasing importance worldwide as a result of the high market demand for caviar, as well as the need to restore the drastic declines in wild sturgeon populations over the past 30 years [1,2]. Sturgeon production constitutes an economically important activity in the aquaculture sector of many European, Asian and American regions [3]. The European Union (EU) stands out as the fifth largest producer in the world, with Spain being the fifth most important producing country within the EU [4]. The Caspian Sea has traditionally been the main place for sturgeon harvesting. This location is known for being the origin of the most valuable caviars, such as the ones obtained from the beluga sturgeon (*Huso huso*), the Danube sturgeon (*Acipenser gueldenstaedtii*) or the starry sturgeon (*Acipenser stellatus*). Nevertheless, the current international market has been reshaped by other sturgeon species of high economic value [4], such as the Siberian sturgeon (*Acipenser baerii*). The long time it takes for female sturgeon to reach sexual maturity, up to six to eight years in captive Siberian sturgeon, becomes the main reason for their high market price [5].

*Aeromonas salmonicida* is a Gram-negative, facultative intracellular bacterium from the *Aeromonadaceae* family [6]. *Aeromonas* is a widespread bacterium and primary non-species-specific pathogen causing furunculosis in fresh and saltwater species [7,8,9,10,11,12]. Lesions of furunculosis and septicaemia caused by *A. salmonicida* have been described in a great variety of fish, causing high mortality outbreaks [13,14,15,16]. Moreover, its zoonotic potential is highlighted by the reports of human infections caused by mesophilic strains [17,18,19,20].

The transmission of *A. salmonicida* is horizontal, either by direct contact with infected fish or indirect contact with fomites, such as microplastics or ectoparasites that feed on fish’s external surfaces [21,22,23]. Several routes of entry have been described in the literature [24,25,26,27]. One possible portal of entry is through the gills, as they are frequently injured by ectoparasites or other pre-existent infections [28]. After its entry, the systemic dissemination from gills is facilitated by the abundant blood capillaries present in this organ. However, cutaneous and digestive routes have been determined as the most frequent ways of infection [24]. Moreover, the mucin composition of the skin or intestine may play a role in the infection, promoting the adhesion of the bacteria [29].

Furunculosis is a multifactorial disease caused by the conjunction of different host and environmental factors [9], and sturgeons are very sensitive to little thermic oscillations [30]. Thus, slight variations in temperature, oxygen concentration and fish stocking density, among others, can lead to an increase in stress on the fish and impair immunity and predispose them to infections and disease [6].

*A. salmonicida* strain pathogenicity, as well as the health status and age of the host, are also factors to be considered in aeromoniasis [6].

Several virulence factors have been characterized for *A. salmonicida*, such as the type 3 secretion system (T3SS), the major virulence system [31]. This key secretion system is commonly found in several Gram-negative bacteria [32,33], allowing toxin translocation from the pathogen to the host [34]. Indeed, although it has been traditionally stated that *Aeromonas* needs a pre-existent lesion to enter the host, *A. salmonicida* itself is able to alter the epithelium integrity, during actin depolymerization induced by T3SS effectors or even by several proteases [35]. Moreover, *Aeromonas* pathogenicity is also driven by the protein structure A layer, which increases its hydrophobic protein surface and favours its adhesion to food debris or animal tissues [36,37].

Moreover, asymptomatic animals may play a key role in the spread of the disease [12,21,37]. Sturgeons act as asymptomatic reservoirs of some pathogens, such as the cyprinid herpesvirus type 3 (e.g., CyHV3) [38]. Therefore, a full understanding of *Aeromonas* spp. pathogenesis in sturgeons may benefit furunculosis control in other farmed fish species.

The present study describes the first natural infection of Siberian sturgeon (*Acipenser baerii*) juveniles by *Aeromonas salmonicida* subsp. *salmonicida*. For this purpose, a detailed macroscopic, histological, immunohistochemical and microbiological study of fish tissue samples obtained during the autopsy was carried out. We anticipate these results could contribute to a better understanding of the pathogenic mechanisms of *A. salmonicida* in sturgeons aside from providing new diagnostic tools for bacterial diseases with an increasing impact on the caviar industry.

## 2. Materials and Methods

### 2.1. Fish and Tissue Sampling

In February 2022, a disease outbreak was recorded in Siberian sturgeons (*Acipenser baerii*) cultured in a local farm in the north of Spain. Four Siberian sturgeon juveniles were sent to the VISAVET Health Surveillance Centre facilities at the Complutense University of Madrid (UCM). Fish were weighed upon arrival and autopsied. During the autopsy, sturgeons were subjected to a detailed macroscopic examination, gills were evaluated for parasites, and tissue samples were collected for subsequent histological and microbiological studies.

### 2.2. Parasitological Study of Gills

For each sturgeon, the operculum was removed, and the gills were extracted. All the arches were collected from each animal and separated individually, numbered from the most external to the inner arch with respect to the operculum.

The gill arch was then cut with a scalpel, and representative samples of gill filaments, including secondary lamellae, were placed on a glass slide with a few droplets of physiological saline solution. Finally, fresh preparations were examined for parasites using brightfield microscopy.

### 2.3. Microbiological Study

During the post-mortem examination, sterile samples of liver, spleen, kidney and telencephalon of each sturgeon were collected in situ and cultured in Columbia, Anacker-Ordal and KDM2 agar plates for bacterial isolation. Then, bacterial isolates were subjected to a protein–peptide extraction protocol based on formic acid–acetonitrile (Bruker Daltonik, Bremen, Germany) to obtain matrix-assisted laser desorption ionization time-of-flight mass spectrometry (MALDI-TOF MS) profiles. Bacterial isolates identified as *Aeromonas* spp. by MALDI Biotyper software (version 3.1; 4613 entries) were subjected to DNA extraction and *Aeromonas salmonicida*-specific PCR protocol [39]. Moreover, the sensitivity of all isolated strains to antimicrobial agents commonly used in aquaculture (florfenicol, flumequine and tetracycline) was studied to rule out the emergence of resistant strains.

In addition, DNA was extracted from the spleen, liver and kidney and conventional PCR targeting of the major capsid protein (MCP) gene of sturgeon iridoviruses was performed using primer sets A, B and D, and the protocol already published by Bigarré and colleagues [40].

### 2.4. Histological Study

Tissue samples were fixed in formalin for 48 h, trimmed and automatically processed (Citadel 2000 Tissue Processor, Thermo Fisher Scientific, Waltham, MA, USA). Then, they were embedded in paraffin and paraffin blocks were formed (HistoStar Embedding Workstation, Thermo Fisher Scientific). Sections of 4 μm thickness for each tissue sample were obtained using a microtome (Finesse ME+ Microtome, Thermo Fisher Scientific) and were automatically stained with haematoxylin and eosin (Gemini AS Automated Slide Stainer, Thermo Fisher Scientific) and Periodic Acid Schiff (PAS) eosin (Gemini AS Automated Slide Stainer, Thermo Fisher Scientific) for their histological evaluation.

### 2.5. Immunohistochemical Study

Paraffin sections were deparaffinised and rehydrated, and a heat-induced epitope retrieval was carried out with a PT module (Epredia; Thermo Fisher; Waltham, MA, USA). Samples were tempered and incubated in a hydrogen peroxide solution in methanol (Panreac; Madrid, Spain) to quench the endogenous peroxidase. Slides were then incubated in horse serum (Vector; Newark, CA, USA), followed by overnight incubation at 4 °C with the primary antibody (mouse monoclonal anti-*Aeromonas* spp. antibody, courtesy of Health Institute Carlos III, Madrid, Spain). Commercial reagents were used for the secondary antibody (ImmPRESS-VR horse anti-rabbit IgG polymer kit; Vector) and chromogen (ImmPACT NovaRED peroxidase substrate; Vector). *Aeromonas* spp. culture-positive rainbow trout cases were used as positive controls. For negative controls, the primary antibody was replaced by a commercial universal negative control reagent. Finally, samples were counterstained with haematoxylin (Gemini AS Automated Slide Stainer, Thermo Fisher) and coverslipped.

## 3. Results

### 3.1. Macroscopic Study

Overall, external examination showed marked size dispersion among sturgeons. Three of them were emaciated (3/4; 75%) (Figure 1a). The average weight was 110.3 g. The gills were diffusely pale pink. Impression smears of the gills did not show any branchial parasites. An increase in pigmentation and mucus production was observed in all the sturgeons (4/4; 100%) (Figure 1b). In the skin, particularly in the dorsal area of the head and the latero-ventral region, there were multifocal irregular haemorrhages with occasional central ulcerations (severe subacute multifocal haemorrhagic and ulcerative dermatitis; 3/4, 75%) (Figure 1c).

The abdominal cavity revealed a slightly enlarged liver (hepatomegaly) with a diffuse pale pink colour and multiple foci of petechial haemorrhages in some fish (2/4; 50%). The spleen also presented an increased size (splenomegaly) (4/4; 100%) and it was diffusely dark red in one of the sturgeons (congestion) (1/4; 25%). The kidney was increased in size and had a diffuse red appearance (congestion) (3/4; 75%) (Figure 1d).

### 3.2. Histological Study

The skin revealed multiple ulcerated regions, being the epidermal layer substituted by an amorphous eosinophilic material admixed with basophilic cellular debris and a severe inflammatory infiltrate, mainly composed of granulocytes and lymphocytes and macrophages (Figure 1a). Moreover, the non-ulcerated areas of the epidermis showed epithelial cells separated by colourless spaces (spongiosis) and a mild thickening of the spinous layer (acanthosis). Mucous cells were increased in number and size (hyperplasia; Figure 2a). The basal layer of the epidermis showed moderate infiltration of inflammatory cells, predominantly lymphocytes, which were diffusely distributed throughout this layer.

The gills showed hyperplasia of the pavement cells and mucous (Goblet) cells (Figure 2c), as well as diffuse expansion of the lamina propria of the gill arch and the filament by a moderate infiltrate of lymphocytes and eosinophilic granulocytes. Occasional atrophy and fusion of secondary lamellae were also observed (Figure 2c).

The lymphoid tissue in the epicardium, spleen and mucosa-associated lymphoid tissue (MALT) revealed a decrease in size and cellularity, as well as multiple lymphoid cells showing karyolysis and karyorrhexis (necrosis) (Figure 2d,e). The liver and kidney were congested, and there were multiple extravasated erythrocytes in the interstitium (haemorrhages). The lamina propria of the intestine was diffusely expanded by a moderate lymphocytic infiltration. The immunohistochemistry revealed positive immunoreaction in the cytoplasm of macrophages located in dermal, splenic, and MALT lesions, as well asthe liver and kidney (Figure 2e,f).

### 3.3. Microbiological Study

The bacteriologic study yielded pure growth of brown-pigmented bacterial colonies. Bacterial isolates were identified to the genus level as *Aeromonas* spp. using MALDI-TOF MS (MALDI-TOF MS-based identification log [score] value, >2.00), and further confirmed as *Aeromonas salmonicida* by PCR [39]. The brown pigmentation of the bacterial colonies is a distinct feature of *A. salmonicida* subsp. *salmonicida* [36].

Antibiogram results showed susceptibility of the bacterial isolates to florfenicol (36–48 mm), flumequine (40–50 mm) and tetracycline (40–42 mm).

None of the DNA samples were amplified with primer sets A, B and D, indicating the absence of iridovirus infection at necropsy. However, the positive control of iridovirus produced a unique signal at the expected size with the three sets of primers.

## 4. Discussion

The lesions and clinical manifestations observed in these juvenile Siberian sturgeons naturally infected by *A. salmonicida* suggest a hyperacute form of the disease. This form is characterised clinically by skin hyperpigmentation, dyspnoea and increased fish mortality. The multiple haemorrhages in several organs, along with the severe ulcerative dermatitis, are typical findings of acute furunculosis [14,28]. Indeed, the haemorrhagic septicaemia observed in these fish specimens is indicative of a hyperacute course of furunculosis [16]. Austin and collaborators described the occurrence of three clinical courses associated with *A. salmonicida* infection, being the hyperacute infection the most common form in young individuals and the chronic course in adults [6].

The histological findings were comparable to the ones described in other fish species suffering from acute aeromoniasis (*A. salmonicida*) (i.e., crucian carp, goldfish, rainbow trout and turbot) [14,41,42]. Moreover, some lesions due to *A. hydrophila* in one Siberian sturgeon case were similar to those described here, as the Siberian sturgeon showed hyperpigmentation and ascites [43]. Skin furuncles caused by *A. salmonicida* have also been reported in the Atlantic Salmon (*A. oxyrinchus oxyrinchus*) [44,45]. The development and progression of *Aeromonas* spp. skin lesions and their clinical manifestations are closely linked to the production of enzymes and toxins, such as gelatinase and lecithinase [11].

The pre-existence of dermal lesions, such as skin abrasions, are predisposing factors for ulcerative dermatitis in fish [46]. Dermal abrasions can be caused by different mechanisms, mainly poor handling of animals and the coexistence of other infectious (viral or bacterial) or parasitic diseases, which were excluded in this study [28]. In addition, the environment constitutes an indispensable element in the development of fish diseases [6]. Sturgeons are very susceptible species to temperature fluctuations, so a deviation outside their physiological range can lead to heat stress and increased morbidity and mortality [47].

Although bacterial colonies in the heart, spleen, liver and kidney are commonly outlined in previous studies [16,28,48], these were not observed in any of our Siberian sturgeon cases infected by *A. salmonicida*. Nevertheless, immunohistochemistry in several organs revealed the presence of *Aeromonas* in the cytoplasm of macrophages. Therefore, immunohistochemistry stands out as a more precise technique to detect *Aeromonas* in tissues, especially in hyperacute cases showing subtle lesions in internal organs.

In fact, we hypothesize that since the sturgeons studied were moribund, there was no time for bacterial colony formation in the tissues. On the other hand, sturgeons are believed to be more resistant to bacterial diseases than other cultured fish, explaining the lower susceptibility of sturgeons to develop severe forms of furunculosis, including septicaemia [3].

Experimental studies of Siberian sturgeon immunization with O-antigen of *A. salmonicida* showed a positive humoral response with specific antibody production and an increase of the phagocytic potential of neutrophils [49,50]. Interleukin 1, 6 and β have been demonstrated to increase with *A. hydrophila* infection in Siberian sturgeon, suggesting the importance of these cytokines in the immune response of the sturgeon and its protection against bacterial infection [51,52,53]. Additional studies regarding sturgeon immune response are required.

*A. salmonicida* is widespread in aquatic environments and infects multiple fish species, causing significant economic losses in aquaculture worldwide. Antimicrobials are the only available treatments so far to control *A. salmonicida* infections. However, some bacterial isolates have undergone resistance to multiple antimicrobials. Previous studies in fish have shown *Aeromonas* spp. resistance to rifampicin, bacitracin and penicillin, among others [42]. Moreover, the ability of *Aeromonas* spp. to form biofilms in the presence of antibiotics leads to treatment failure and infection recurrence [54]. Here, the antibiogram revealed that the *A. salmonicida* strain isolated from diseased juvenile sturgeons was susceptible to three common antimicrobials used in aquaculture. The potential reservoir role of sturgeons in some notifiable diseases of fish has been suggested [38], but their particular role in the inter- and intra-species epidemiology of furunculosis is yet to be defined.

To the best of our knowledge, this is the first description of natural infection by *A. salmonicida* subsp. *salmonicida* in Siberian sturgeons. The results shown here provide valuable information regarding the pathogenicity and susceptibility of *A: salmonicida* subsp. *salmonicida* isolates in new aquaculture-rearing species such as the sturgeon. The ability of *Aeromonas* spp. to survive in a wide variety of environments, aside from its zoonotic potential, makes this pathogen a public health concern worldwide. Future studies may increase the overall understanding of the pathogenicity and susceptibility of *Aeromonas* spp. and the role of the sturgeon in the transmission of the disease.

## 5. Conclusions

Juvenil sturgeons suffere from a hyperacute form of the disease;The *A. salmonicida* subsp. *salmonicida* strain isolated was susceptible to common antimicrobials used in aquaculture;Bacterial colonies were not seen histologically, but immunohistochemistry has been useful for detecting *Aeromonas* in tissues.

## Figures and Tables

**Figure 1 animals-13-02697-f001:**
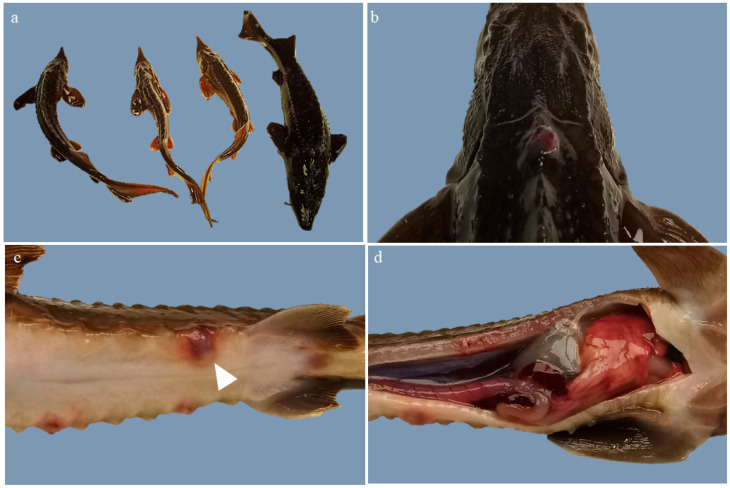
Gross lesions in Siberian sturgeons (*Acipenser baerii*) naturally infected with *Aeromonas salmonicida*. (**a**) Marked size heterogeneity; (**b**) Skin. Increased mucus production and hyperpigmentation; (**c**) Skin. Severe, multifocal, haemorrhagic and ulcerative dermatitis (arrow head); (**d**) Abdominal cavity. Ascites, hepatomegaly and multiple foci of petechial haemorrhages, haemorrhages, splenomegaly and organic congestion.

**Figure 2 animals-13-02697-f002:**
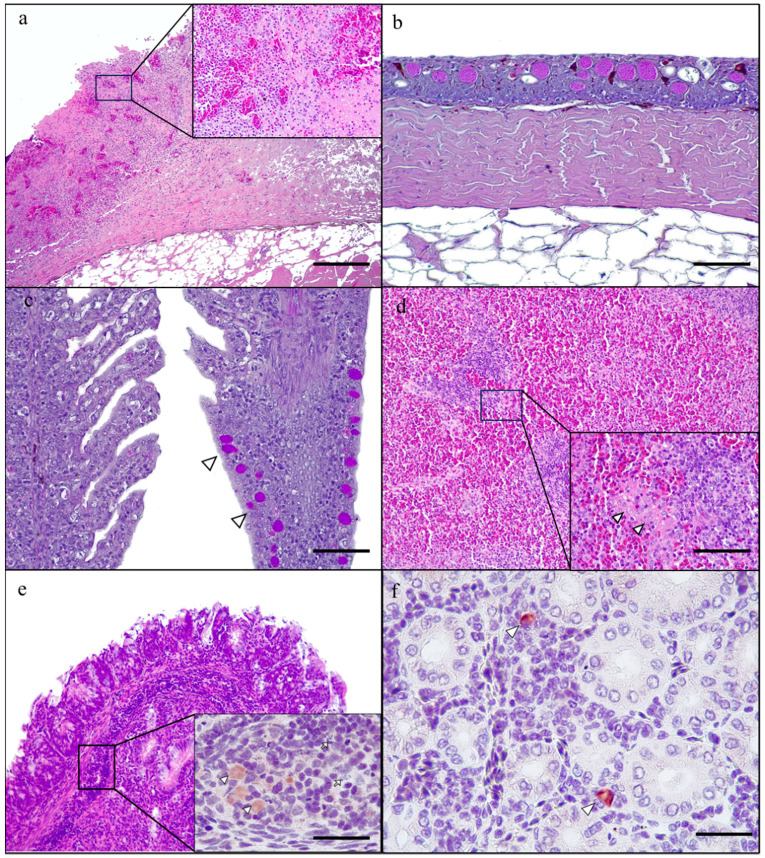
Histological findings in Siberian sturgeons (*Acipenser baerii*) infected with *Aeromonas salmonicida*. (**a**) Skin. Severe, acute, ulcerative dermatitis. Inset: necrosis and haemorrhages admixed with numerous cellular debris and granulocytes (H&E; scale bar: 500 μm); (**b**) Skin. Goblet cells hyperplasia (PAS; scale bar: 50 μm); (**c**) Gills. Right gill shows atrophy and fusion of the secondary lamellae and mucous cell hyperplasia (arrowhead). Less affected gill at the left. (PAS; scale bar: 50 μm); (**d**) Spleen. Lymphoid tissue necrosis. Inset: necrosis (arrow head) at higher magnification (H&E; scale bar: 200 μm); (**d**,**e**) Intestine. Mucosal-associated lymphoid tissue (MALT) necrosis (H&E; scale bar: 50 μm). Inset: macrophages show positive immunoreaction against *Aeromonas* (arrow heads), and there is karyolysis and karyorrhexis of lymphoid cells (necrosis) (arrows) (mouse monoclonal anti-*Aeromonas* spp. antibody; scale bar: 500 μm). (**f**) Kidney. There are scattered macrophages showing positive immunoreaction against *Aeromonas* (arrowhead) in minimal interstitial nephritis (mouse monoclonal anti-*Aeromonas* spp. antibody; scale bar: 200 μm).

## Data Availability

The data that support the findings of this study are available from the corresponding author upon reasonable request.

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
