# Peer review of "An Outbreak of Aeromonas salmonicida in Juvenile Siberian Sturgeons (Acipenser baerii)"

_animals, 2023, doi:10.3390/ani13172697_

Round 1

Reviewer 1 Report

There are several points in the Introduction section with outdated information and misinterpreting some authors regarding the state of the art on A. salmonicida. This should be thoroughly reviewed and corrected to provide an adequate description of the bacterium and its pathogenesis.

This reviewer disagrees with some of the gross diagnoses made by the authors. Similarly, this reviewer believes that there are some microscopic lesions that have been misinterpreted by the authors. Therefore, the text must be modified according to the images or provide new images that support the statements.

There are also multiple grammatical errors that need to be corrected before the publication of this document.

A point by point revised MS can be found in the attached PDF.

Reviewer 2 Report

This is a useful manuscript, documenting a case of Aeromonas in sturgeon and comparing/contrasting the findings with similar etiological-based cases in other fish species.    However, I did find that the histopathology section would benefit from improvement.  The gross findings of ulceration and appearance of septicemia were not well noted within the histopathology.  The section of skin was not of an ulcerated area and therefore not really germane to the clinical case.  A section from an ulcer and immediately adjacent to the ulcer is more likely to show the nature of the host-pathogen interaction and likely reveal the presence of bacteria.  This would make a much stronger case for the overall conclusion of cause-effect.  The gill lesions are vague and non-specific.  Authors conclude that pavement (lamellar epithelial cells) have increased in number, but really what they show is evidence of lamellae being covered by a more cuboidal type of epithelial cell.  This might reflect an augmented chloride cell population (perhaps a metaplastic response).  Showing normal gill alongside would be useful.   Figure 2 e, does not show many of the renal lesions described in the paper.  The eosinophilic material in the tubules is equivocal, and may simply be exfoliation of apical parts of the renal tubular epithelial cells (not a cast...rather an early artifact from fixative not penetrating quickly enough).   Figure 2 d is confusing.  Part of what is shown looks like red blood cell nuclei that have exsheathed from a clump of red blood cells.   Overall, a stronger presentation of histopathology is needed to complete this case report;  the lesions shown should correspond with the descriptions elsewhere in the paper.   At the moment, the histology is not supportive of the diagnosis.

Round 2

Reviewer 1 Report

I would like to thank the authors for their work to modify the MS according to the reviewer's suggestions. Now, I find the MS meets the requirements to be published in Animals.

There are still some gramathical mistakes that will be easily corrected by authors when they review the MS.